# Multi-Hazard Susceptibility Assessment Using the Analytical Hierarchy Process in Coastal Regions of South Aegean Volcanic Arc Islands

Pavlos Krassakis [1,2,*], Andreas Karavias [2], Paraskevi Nomikou [3], Konstantinos Karantzalos [4], Nikolaos Koukouzas [2], Ioannis Athinelis [1], Stavroula Kazana [3] and Issaak Parcharidis [1]

1   Department of Geography, Harokopio University of Athens, El. Venizelou 70, 17671 Athens, Greece
2   Centre for Research & Technology Hellas (CERTH), 15125 Athens, Greece
3   Department of Geology and Geoenvironment, National and Kapodistrian University of Athens, Panepistimioupoli Zografou, 15784 Athens, Greece
4   Department of Topography, School of Rural and Surveying Engineering, National Technical University of Athens, Zografou Campus, 9 Heroon Polytechniou Str., 15780 Athens, Greece
*   Correspondence: krassakis@certh.gr

**Abstract:** Coastal environments are highly recognized for their spectacular morphological features and economic activities, such as agriculture, maritime traffic, fishing, and tourism. In the context of climate change and the evolution of physical processes, the occurrence of intense natural phenomena adjacent to populated coastal areas may result in natural hazards, causing human and/or structural losses. As an outcome, scientific interest in researching and assessing multi-hazard susceptibility techniques has increased rapidly in an effort to better understand spatial patterns that are threatening coastal exposed elements, with or without temporal coincidence. The islands of Milos and Thira (Santorini Island) in Greece are prone to natural hazards due to their unique volcano-tectonic setting, the high number of tourist visits annually, and the unplanned expansion of urban fabric within the boundaries of the low-lying coastal zone. The main goal of this research is to analyze the onshore coastal terrain's susceptibility to natural hazards, identifying regions that are vulnerable to soil erosion, torrential flooding, landslides and tsunamis. Therefore, the objective of this work is the development of a multi-hazard approach to the South Aegean Volcanic Arc (SAVA) islands, integrating them into a superimposed susceptibility map utilizing Multi-Criteria Decision-Making (MCDM) analysis. The illustrated geospatial workflow introduces a promising multi-hazard tool that can be implemented in low-lying coastal regions globally, regardless of their morphometric and manmade characteristics. Consequently, findings indicated that more than 30% of built-up areas, 20% of the transportation network, and 50% of seaports are within the high and very high susceptible zones, in terms of the Extended Low Elevation Coastal Zone (ELECZ). Coastal managers and decision-makers must develop a strategic plan in order to minimize potential economic and natural losses, private property damage, and tourism infrastructure degradation from potential inundation and erosion occurrences, which are likely to increase in the foreseeable future.

**Keywords:** multi-hazard susceptibility assessment; coastal zone; South Aegean Volcanic Arc; GIS; RES; AHP

## 1. Introduction

Southern European countries, and their low-lying coastal areas, offer Sun, Sea, and Sand (3S), responsible for a significant portion on their total revenues [1]. Coastal tourism is one of the most important economic sectors in Greece, where it accounts for 13% of the country's Gross Value Added (GVA) and 3.8% of employment [2]. Regarding the Greek Population-Housing Census 2021, the South Aegean territory was the only region in Greece that recorded a population increase by 5% [3] from 2011 to 2021. Therefore, the SAVA

islands are a hot spot for anthropogenic activities, particularly during the tourist season, which starts in April and continues through the end of October each year.

In terms of natural environment, the SAVA area is one of the most seismically and volcanically active areas in Europe, serving as a one-of-a-kind physical laboratory for scientists and visitors. The Hellenic Subducting System (HSS) and regional tectonics have produced several volcanic edifices such as Methana, Milos, Santorini and Nisyros, which are strongly associated with the area's recent geological history [4].

According to the report of the Intergovernmental Panel on Climate Change (IPCC), [5] the Aegean Sea will be one of the most vulnerable areas in the Mediterranean basin to global warming and climate change scenarios, with more extreme meteorological events, heat waves and droughts. The resilience of coastal infrastructure and topography are key parameters in mitigating susceptibility to torrential floods, landslides, tsunamis, and Sea Level Rise (SLR) [6]. According to Krassakis, et al. [7], low-lying areas, in terms of the Extended Low Elevation Coastal Zone (ELECZ), require holistic coastal planning, due to their high urbanization expansion and soil loss.

Up to the present time, a great number of investigators have focused on single risks with multiple moderating factors [8–10]. While susceptibility modeling approaches for single processes have advanced considerably, there is neither a standard terminology nor a systematic conceptual approach for analyzing multiple hazards in combination, not surprisingly, since multi-hazard analyses do not account for the total number of single-hazard assessments [11,12]. However, evaluating multiple hazards and their multiple controlling factors is a complex issue.

The term generally used, "multi-hazard", indicates all the major hazards that are present in a specific geographic area [13]. The relationships between the various types of hazards may range from interactions to cascades and domino effects [14–16]. It is worth noting that many regions of the world may experience multiple hazards simultaneously [17–19]. The same logic can be adopted also for coastal regions that are susceptible to different types of hazards, any one of which may manifest individually or in conjunction [20–24]. From this perspective, the SAVA islands may be susceptible to different hazards such as: (1) soil loss, (2) landslides, (3) torrential floods, and (4) tsunamis, which can endanger low-lying coastal areas unexpectedly [25–29]. Adopting a multi-hazard scenario in terms of the simultaneous occurrence of the investigated hazards could identify the most vulnerable hot-spot areas in Milos and Thira.

Historically, Thira and Milos have suffered frequently from landslide events, causing serious damage along transportation networks, seaports, pathways and beaches near to cliffs. According to the records, landslides and/or extreme rainfall events have occurred in Thira Island, in Oia, 2011, Fira, 2022, Red beach, 2018, etc., and in Milos Island, in 1992 Palaiochori and Firiplaka, and in 2017 Papafragas beach. In addition, intense weather fronts named 'Eva', 'Elpida' 'Ballos' etc., have become more frequent in Greece and by extension in the Cyclades, putting lives and infrastructure in danger. Therefore, soil loss and erosion need to be monitored frequently, especially due to the unplanned urbanization expansion adjacent to the existing drainage networks and coastal cliffs, which are connected with steep and increasing slopes and terrain [30–33].

Aside from hydrometeorological hazards, in 1956 the fault zone of Santorini–Amorgos produced the largest tsunami that year, followed by a seismic event with a magnitude of Mw = 7.4. According to the literature, the developed seismo-genic tsunami recorded a run-up of at least 10 m above sea level (a.s.l.) [34–38] along coastal areas of the Cyclades complex [39].

Taking into consideration all the above mentioned hazards, this work introduces an innovative approach that could act as a baseline for multi-hazard approaches and early warning systems in the wider area of the Aegean Sea. This manuscript presents a methodology for integrating and analyzing multiple hazards, in the same geographic area with similar morphological and geological characteristics. Ultimately, the findings of this research could contribute to the effective and holistic management of low-lying coastal

regions in the context of climate change adaptation, mitigation strategies, and multi-hazard assessment. In terms of geospatial data, this work incorporates for the first-time public road data from Microsoft Bing Maps, combined with the Open Street Map (OSM) data, in order to enhance the accuracy of the transportation network.

## 2. Study Areas

The South Aegean Region extends from 24°19′ E, 36°42′ N to 25°29′ E, 36°22′ N, where the volcanic islands of Thira (Santorini Island complex) and Milos are located.

The South Aegean Volcanic Arc islands (SAVA) are distinguished by both subaerial and submerged volcanism and starts from the Saronic Gulf in the west to the Nisyros volcanic field (Figure 1) in the east [40]. The SAVA has been developed over the past 5 million years in the Hellenic Subduction System's (HSS) continental crust due to the northward subduction of the last remnant of the African plate's oceanic crust beneath the active margin of the European plate [41]. Milos and Thira's geological context is dominated by volcanic or volcano-sedimentary rocks, including lavas, dykes, pumice, tuffs, and scoria.

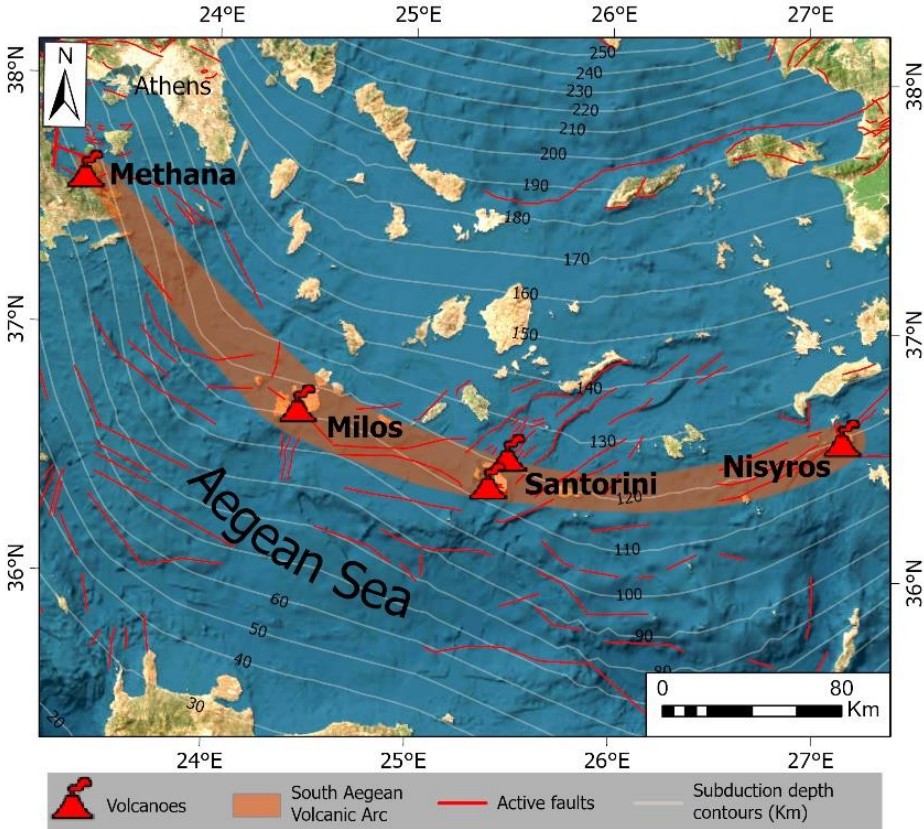

**Figure 1.** Synthetic topographic Map of the South Aegean Volcanic Arc (SAVA) islands. The SAVA zone is depicted in orange, the volcanoes in red, and the subduction depth contours (km) in white. Basemap source: Esri World Imagery.

## 3. Materials and Methods

### 3.1. Data Collection

The datasets used for the processing of this study were based on open access products from the Copernicus program and other open-source databases, which are described in the Table 1. Specifically, geological information was obtained from the Hellenic Survey of Geology and Mineral Exploration (HSGME). Additionally, the land use data were processed from the Corine 2018 dataset. The Digital Elevation Model (DEM) was acquired from the National Cadastre in order to extract the required morphological factors for the multi-hazard

analysis (Figures 2 and 3). The urban fabric was obtained from previous study [7] and the transportation network from the OpenStreetMap (OSM) and Microsoft [42]. Moreover, Sentinel 2 satellite images were utilized for the calculation of the Normalized Difference Vegetation Index (NDVI) and Bare Soil Index (BSI).

**Table 1.** Datasets used for multi-hazard modelling.

| Data Usage | Data Source | Spatial Resolution/Scale | Temporal Scale | Primary Format |
|---|---|---|---|---|
| NDVI & BSI | Sentinel-2 [43] | 10 m | 2016–2021 | Raster |
| DEM | National Cadastre | 5 m | 2019 | Raster |
| Road network | Open Street Map/Microsoft [44] | - | 2016–2022 | Vector (polylines) |
| Urban fabric and coastline | Krassakis et al. [7] | | | Vector (polygons/polyline) |
| Pan-European/CORINE Land Cover/CLC 2018 | Copernicus–EU [45] | 100 m | 2018 | Vector (polygons) |
| Geology of Milos and Thira islands | HSGME [46,47] | 1:50,000 | 1977, 1980 | Raster |
| Beaches, ports, & airports | ESRI basemap | 1:10,000 | 2021 | Vector (polygons/polylines) |
| Active Faults | National Observatory of Athens [48] | - | 2019 | Vector (polylines) |
| Earthquake epicenters | National and Kapodistrian University of Athens [49] | - | 1900–2020 | Vector (points) |
| Subduction depth contours | European Commission [50] | - | | Vector (polylines) |
| Precipitation (mm) | Hellenic National meteorological service (HNMS) [51] | - | 1971–2020 | Raster (grid) |
| K-factor | Joint Research Centre (JRC) [52] | 500 m | 2015 | Raster (grid) |

### 3.2. Methodology

According to the following workflow (Figure 4), a fourfold process was incorporated: (1) the development of a geodatabase; (2) the generation of the multiple hazards individually in terms of landslide, torrential flood, soil erosion and tsunami; (3) the visualization and classification of the selected criteria for each hazard; and (4) the multi-criteria ranking of the derived hazard models utilizing, the Analytical Hierarchy Process (AHP) methodology. The first step was heads-up digitization utilizing high resolution ESRI imagery in an attempt to delineate the bounds of important infrastructure and beaches surrounding the two islands. After that, multispectral Sentinel-2 pictures from 2016 and 2021 were obtained [7] in order to generate the Normalized Difference Vegetation Index (NDVI) and Bare Soil Index (BSI), which are required for soil erosion and torrential flooding approaches.

The components depicted in the right portion of the workflow were incorporated into the relational database, and converted into $10 \times 10$ m raster datasets. Thus, the findings from each individual hazard model were estimated over the total extent for both islands and then subtracted for the low-lying coastline zone.

In accordance with the applied Multi Criteria Decision Making (MCDM) [53] approach, the AHP was adopted, considering the frequency of the potential natural hazards in Milos and Thira Islands. For the implementation of the AHP approach, all the criteria were reclassified into a single tactical scale from 1 to 5, in order to be comparable to each other and be homogenized. The classification of each criterion was implemented using the natural breaks algorithm (Jenks) method [54], which is one of the most appropriate methods for classification [55].

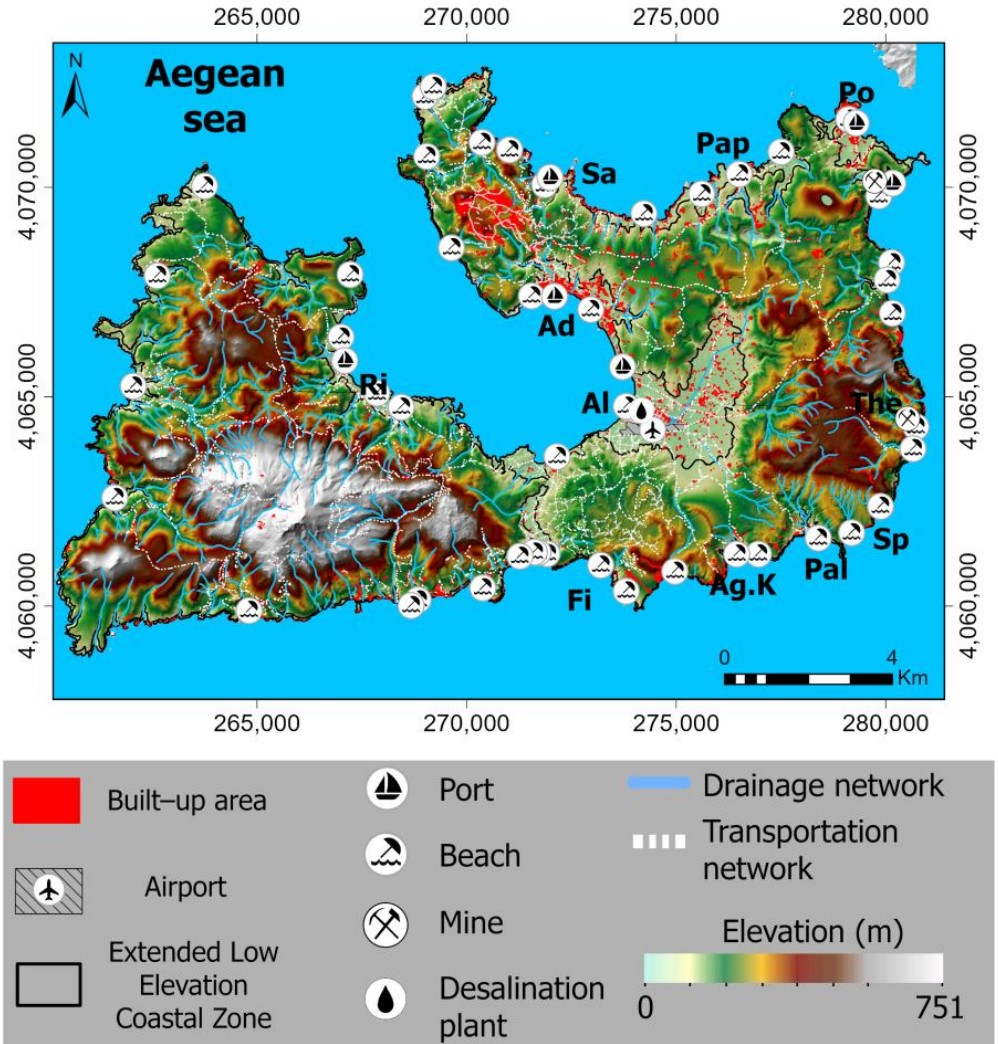

**Figure 2.** Digital Elevation Map of Milos Island illustrating the input of geospatial analysis, such as seaports, beaches, desalination plant and mines. Ad: Adamas, Al: Alikes, Po: Posidonia, Ri: Rivari Lagoon, Pal: Palaiochori, Pap: Papafragas, Fi: Firiplaka, Sa: Sarakiniko, Sp: Spathi, Th: Thiorichio and Ag.K: Agia Kiraki.

*3.3. Landslide Susceptibility*

    Mapping areas susceptible to landslides is an important tool for disaster management and development activities, such as the planning of infrastructure, settlement, and agricultural activities. In this study, the Rock Engineering System (RES) method was implemented [56], which utilizes an Interaction matrix to determine the interactions between different factors, based on their importance. The RES method is a semi-quantitative method usually applied to problems concerning rock engineering by determining the process through which a given factor causes the problem that is being researched [57–61]. Specifically, this is achieved by studying the way any given causal factor interacts with another in a clockwise manner (Figure 5).

    Furthermore, this method utilizes an Interaction Matrix (see Section 4.1), to determine the degree of interaction between pairs of causal factors. An interaction value of 1 represents no causal interaction between the two factors, while the higher the value, the easier that one of the factors can affect the other.

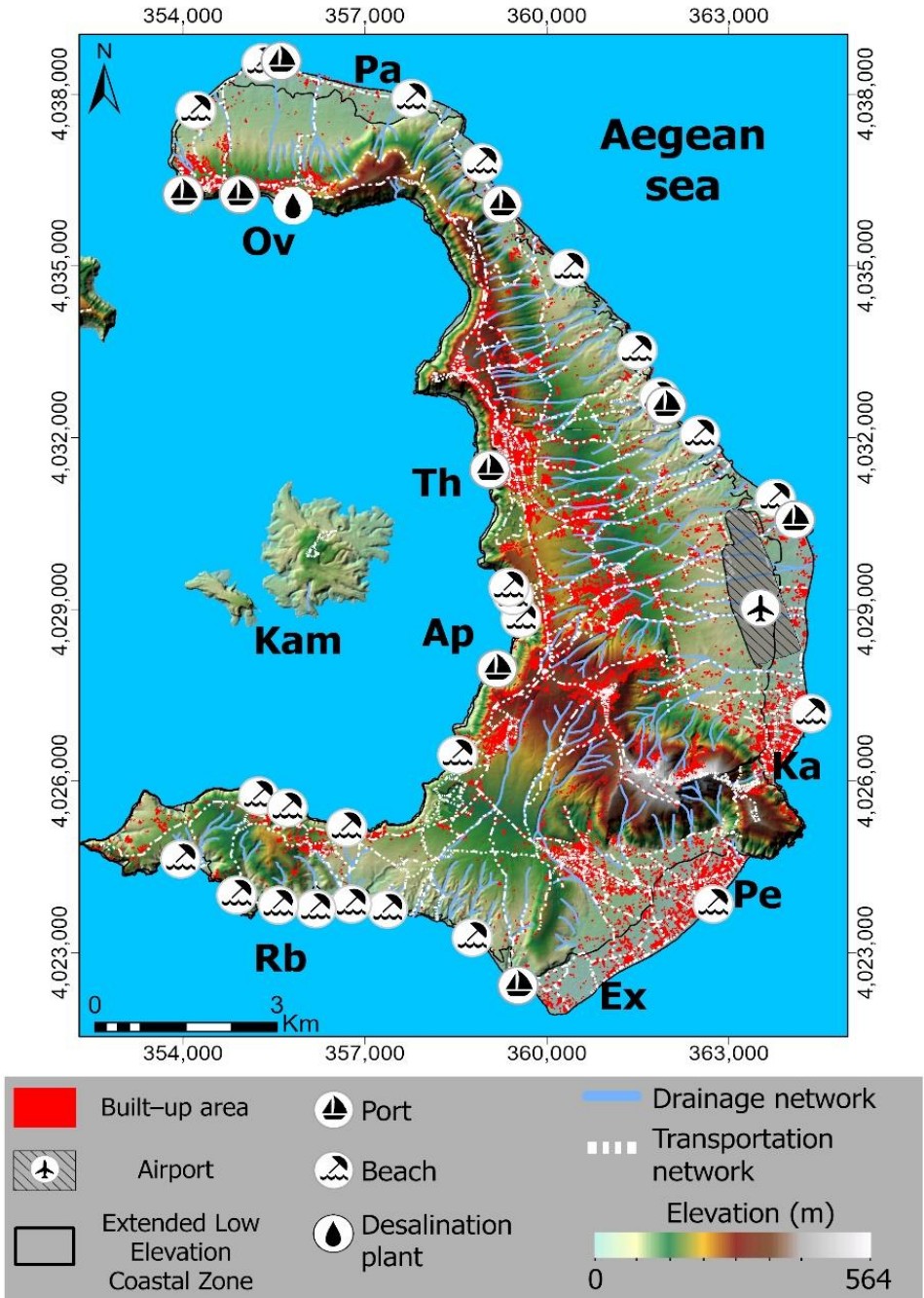

**Figure 3.** Map of Thira Island illustrating the input of geospatial analysis, such as seaports, beaches, desalination plant and mines. Ka: Kamari, Pe: Perissa, Ap: Athinios, Rb: Red beach, Ex: Exomitis, Th: Thira, Pa: Paradisos, Ov: Oia and Kam: Kameni.

Conditioning Factors

The following factors were selected for Milos and Thira: the lithology, the elevation, the slope gradient and curvature, the annual solar radiation, the proximities to tectonic structures, drainage and transportation networks, seismic parameters, and the mean annual precipitation.

- *Lithology*

Lithological information is one of the most critical factors influencing an increase in the likelihood of a landslide occurrence. Depending on geotechnical, stratigraphic, mechanical or hydrogeological characteristics, it is directly related to a slope's resistance to landslide

events [60,62]. Consequently, formations such as flysch are mechanically more unstable and prone to trigger landslide phenomena. Likewise, carbonate rocks, such as limestones, are capable of containing a great amount of water and are very susceptible to dissolution by water and by extensive landslide events. In our case, lavas, dykes and pumice were considered as volcanic rocks, while tuffs and scoriae were grouped as volcano-sedimentary rocks. The Quaternary and more recent sediments for both islands were generally identified as "loose sediments", and scree and mudflows as "Scree & debris flows" (Table 2).

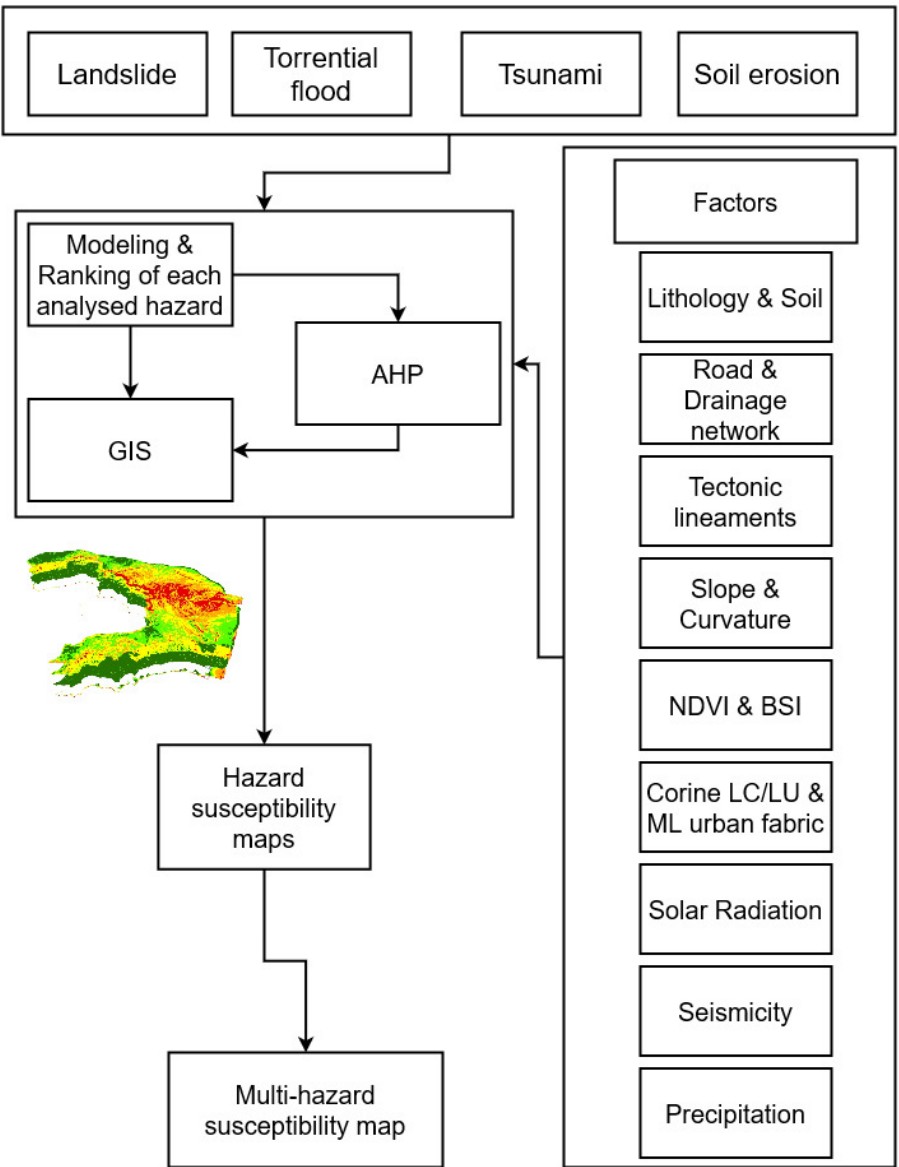

**Figure 4.** Schematic workflow of the implemented multi-hazard approach.

**Table 2.** Landslide causal factors classification and ranks in Milos and Thira island.

| Causal Factor | Class | Rank Values |
|---|---|---|
| | Debris flows & Scree | 5 |
| | Schists & Volcanic rocks | 4 |
| Lithology | Volcano-sedimentary rocks | 3 |
| | Sedimentary rocks & Limestones | 2 |
| | Loose Sediments | 1 |

**Table 2.** *Cont.*

| Causal Factor | Class | Rank Values |
|---|---|---|
| Slope Gradient (°) | >40.40 | 5 |
| | 27.71–40.39 | 4 |
| | 16.45–27.70 | 3 |
| | 7.51–16.44 | 2 |
| | <7.50 | 1 |
| Curvature | <−18.72 | 5 |
| | −14.78−−3.95 | 4 |
| | −3.95−−0.01 | 3 |
| | >0.01 | 2 |
| | 0.01−−0.01 | 1 |
| Mean Annual Rainfall (mm) | >431.06 | 5 |
| | 422.89–431.05 | 4 |
| | 414.09–422.88 | 3 |
| | 406.08–414.08 | 2 |
| | <406.07 | 1 |
| Distance from the river network (m) | <149.33 | 5 |
| | 149.34–355.56 | 4 |
| | 355.57–640.00 | 3 |
| | 640.00–1073.78 | 2 |
| | >1073.79 | 1 |
| Distance from the road network (m) | <132.29 | 5 |
| | 132.30–321.28 | 4 |
| | 321.29–566.96 | 3 |
| | 566.97–894.54 | 2 |
| | >894.55 | 1 |
| Distance from tectonic structures (m) | <86.59 | 5 |
| | 86.60–188.24 | 4 |
| | 188.25–308.70 | 3 |
| | 308.71–474.35 | 2 |
| | >474.36 | 1 |
| Annual Solar Radiation (Wh/m$^2$) | >50,094.64 | 5 |
| | 42,347.82–50,094.64 | 4 |
| | 35,350.69–42,347.82 | 3 |
| | 26,104.48–35,350.69 | 2 |
| | <26,104.48 | 1 |
| Earthquake Kernel Density (magnitude/m$^2$) | >0.14 | 5 |
| | 0.11–0.14 | 4 |
| | 0.07–0.11 | 3 |
| | 0.04–0.07 | 2 |
| | <0.04 | 1 |
| Earthquake Depth (km) | <11.52 | 5 |
| | 11.52–16.88 | 4 |
| | 16.88–21.61 | 3 |
| | 21.61–27.44 | 2 |
| | >27.44 | 1 |

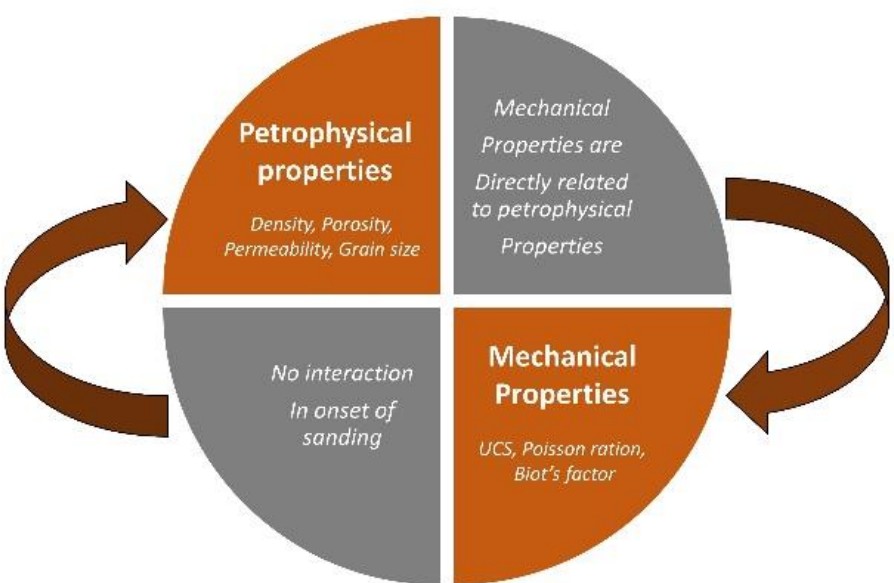

**Figure 5.** Interaction logic between two factors [54].

- *Slope gradient*

Numerous studies have shown that areas with steeper slopes and a more distinct topography are significantly more susceptible to landslides. Therefore, the steeper the slope, the greater the likelihood of a landslide [63–65]. The slope gradient layer was derived from a resampled DEM with a $10 \times 10$ m$^2$ cell size, categorized into five classes (Table 2).

- *Slope Curvature*

The shape of an area's slopes can control the water accumulation, thus affecting their landslide susceptibility. Shape can be classified into concave or convex, where concave slopes tend to hold more water, thus being more saturated and more unstable than convex slopes. [63,66]. This makes concave slopes more likely to cause a landslide. In general, the more negative a value, the steeper the slope, whereas the closer a value is to 0, the flatter the surface (Table 2).

- *Annual Solar Radiation (WH/m2)*

Areas exposed to solar radiation for extended periods of time annually tend to lose much of their moisture quickly. This creates porous gaps between the rock's grains which contain no water, which would normally widen them, thus decreasing the slopes stability. Therefore, slopes with high annual solar radiation exposure are less susceptible to landslides [67]. The solar radiation layer was categorized into five classes (Table 2).

- *Proximity to tectonic structures (m)*

In most instances, tectonic fabric is directly related to slope failure. Specifically, slopes close to faults are more likely to be affected and mechanically weakened, resulting in a landslide. Landslides are more likely to affect regions close to tectonic structures; therefore, it is essential to analyze this factor. Additionally, a raster of Euclidean distances to faults was calculated, which was then classified into five classes (Table 2).

- *Distance from the main road network*

For the creation of distance from the road network, a shapefile of the main road network, combing OSM and Microsoft datasets for the two islands, was utilized, using Euclidean distance. The resultant distance raster was reclassified into five classes (Table 2). It is important to underline that, when a road is blocked by a landslide event, the entire network is affected, as transportation becomes more difficult, especially in islands, where the road network is limited.

- *Mean annual precipitation*

From 1971 to 2022, precipitation data were collected from the Climate Atlas of Greece provided by the National Meteorological Service, in order to create a rainfall map. It should be noted that, although precipitation heights are low, a small amount of rain can be sufficient to trigger landslides if the duration of rainfall exceeds a certain threshold [68]. The generated dataset was also divided also into five classes (Table 2).

- *Seismic parameters*

During an earthquake, a massive amount of stress is caused to the geological formations of the area, thus reducing their stability, leading to landslide events. Since an earthquake can make an area highly unstable [69], a high earthquake density is going to increase the susceptibility of the area drastically. To calculate the earthquake density, an earthquake epicenter inventory for magnitudes greater than 3.5 was collected from the National and Kapodistrian University of Athens from 1900 to 2022, and classified into five categories (Table 2).

The second seismic parameter considered for landslide susceptibility was earthquake depth [69]. Lower earthquake depths tend to cause a greater negative effect on slope stability. As a result, an earthquake depth layer was created, separated into five classes (Table 2).

*3.4. FFPI Methodology*

In accordance with the study conducted by Durlević et al. [70], the Flash Flood Potential Index (FFPI) was adopted in an attempt to assess the terrain susceptibility to torrential flooding. This index represents the probability that a flash flood may occur in a specific research region, illustrating its susceptibility and vulnerability to this hazard. This methodology is based on four conditioning factors, in terms of their effect on surface runoff: the terrain slope, the land cover, the soil type and the vegetation density [71,72]. These datasets were classified based on runoff and flash flood potential and assigned relative *FFPI* ratings using the following Equation (1):

$$FFPI = \frac{S + LC + ST + V}{4} \tag{1}$$

where *S* is the Slope, *LC* is the Land Cover, *ST* is the Soil Type, *V* is the Vegetation density and the variations of n are the respective weights of each factor.

Torrential Flood Factors

Torrential floods and flash floods are natural hazards of great complexity. In a flood-based analysis, the parameters regarding the topography of the study area are considered to be the most crucial.

- *Slope gradient*

Among all of the potential causes of flash floods, topographical gradient is a key parameter, due to the fact that a steep slope, depending on the angle, can create surface water runoff at higher speeds, leading to high water accumulation at the base of those slopes. Consequently, this increases the likelihood of a flash flood occurring significantly, yet the slopes themselves do not typically become flooded. Instead, the low-lying areas that are located at the bottom of the slope are those that are confronted with the greater repercussions of a potential flash flood. As a result, the locations that have a low slope gradient were assigned with a greater relative FFPI in this study.

When determining the FFPI of the slope gradient, the method of Minea et al. [73] was considered, where slope values were calculated in degrees and then reclassified in five classes, as presented in Table 3.

- *Soil type*

The type of an area's soil is considered as responsible for its ability to either drain the surface water, or allow it to move as runoff. Soils such as clays have extremely low permeability and cause increased runoff, resulting in flash flooding. However, soils such as sands are highly permeable, due to larger gaps between the grains, thus being more easily infiltrated by surface water, and preventing flash flood events [74].

The mechanical and hydrological properties of the soil type of a study area control its susceptibility to torrential floods in different ways. Due to the lack of soil maps, lithological datasets were used in an attempt to convert the lithology into soil categories. The correlation between the simplified lithologies and the different soil types was based on research. For example, volcanic or volcano-sedimentary rocks were considered as Andosols, loose sediments and deposit varieties were classified as Alluvials, carbonate rocks were connected to Regosols, and sedimentary rocks, such as sandstones, were considered as Cambisols. Specifically, in this work, the Alluvials are considered the most susceptible soil type, while andosols present the lowest susceptibility, due to increased runoff. As a result, the soil type was also divided into five distinct groups (Table 3).

- *Land cover/Land use*

The use of land can affect the ability of the ground surface to contain water or cause runoff, similar to the effect of soil type [75–77]. Anthropogenic artificial surfaces such as roads, ports or airfields can increase the impermeability of the ground significantly, thus causing more runoff and allowing easier flooding [70,74]. Likewise, areas with little to no vegetation have reduced ability to absorb water, as runoff travels at extreme speeds, thus causing flash floods before it can infiltrate the ground. Vegetated regions, on the other hand, have a natural slowing effect on surface water flow, due to the fact that they allow water ground infiltration, preventing flooding. The necessary data were collected from Corine (2018) (Table 3).

- *Vegetation density*

One of the most important factors that may prevent the occurrence of a flash flood is the existence of vegetation [78]. Particularly, trees and forests, which are known for their stability, can decrease the rate of surface water runoff when they come into contact with it. This becomes even more apparent in areas where there is a higher density of plants and trees [76]. The vegetation density layer was created using the method suggested by [70], in which the Bare Soil Index (BSI) of the study area is calculated in order to determine the areas covered by vegetation by separating the bare areas [79]. This was accomplished using satellite images derived from Sentinel 2, from 2016 and 2021, using the following Equation (2):

$$BSI = \frac{(B11 + B4) - (B8 + B2)}{(B11 + B4) + (B8 + B2)} \tag{2}$$

where *B*11 is the Short-wave infrared band, *B*4 is the Red band, *B*8 is the Near Infrared band and *B*2 is the Green band of the Sentinel-2 satellite system. The following Equation, suggested by [80], was utilized to calculate the vegetation density (*V*) based on the *BSI* index.

$$V = 7.68 \times ln(BSI + 1) + 8 \tag{3}$$

The results were categorized into five classes, as represented in the following Table.

**Table 3.** Torrential flood causal factor classification and ranks in Milos and Thira islands.

| Causal Factor | Class | Rank Values |
|---|---|---|
| Slope Gradient (°) | <7.50 | 5 |
| | 7.51–16.44 | 4 |
| | 16.45–27.70 | 3 |
| | 27.71–40.39 | 2 |
| | >40.40 | 1 |
| Vegetation Density | <7.43 | 5 |
| | 7.43–8.04 | 4 |
| | 8.04–8.45 | 3 |
| | 8.45–9.46 | 2 |
| | >9.46 | 1 |
| Soil Type | Alluvials | 5 |
| | Andosols | 3 |
| | Cambisol | 2 |
| | Regosols | 1 |
| Land cover/Land use | Airports & Ports Mineral extraction sites Non-irrigated arable land Sparsely vegetated areas Salines | 5 |
| | Complex cultivation patterns & Pastures Land principally occupied by agriculture, with significant areas of natural vegetation Bare rocks | 4 |
| | Olive groves Natural grasslands | 3 |
| | Continuous urban fabric & Discontinuous urban fabric | 2 |
| | Sclerophyllous vegetation | 1 |

*3.5. Soil Loss Susceptibility Assessment–Revised Universal Soil Loss Equation (RUSLE)*

A great variety of soil erosion models have been developed over the years, used in the estimation of soil loss and its impact, along with the increase in land conservation efficiency. One of the most commonly utilized methods of soil erosion rate prediction is the Revised Universal Soil Loss Equation (RUSLE), first published in 1991, a modified version of the original Universal Soil Loss Equation (USLE). The main difference and advantage of RUSLE is the ability to estimate the soil loss caused by both surface and rill erosion [81–86]. The RUSLE model estimates the (annual) average soil loss (*A*) in tons per hectare per year (t ha$^{-1}$ y$^{-1}$), based on six major numerical factors (Equation (4)):

$$A = R \times K \times LS \times C \times P \tag{4}$$

where *R* is the rainfall erosivity, *K* is the soil erodibility, *LS* is a factor that usually combines the length (**L**) and the steepness (**S**) of slope, *C* expresses the land cover and *P* is the erosion control practice factor [85].

RUSLE Factors

Soil erosion is a natural hazard like many others, but unlike other risks, it is a dependent variable, making investigation challenging. To evaluate the vulnerability of Milos and Thira to soil loss, the following factors were investigated.

- *Rainfall (R) factor*

The R factor in the RUSLE equation is used to determine the kinetic energy of the rainfall intensity [87,88]. An empirical Equation was implemented due to the lack of intensity data availability, in order to calculate the R factor.

$$R = -8.12 + 0.562 \times P \tag{5}$$

where *P* is the mean annual rainfall in mm of the two study areas. This equation requires long-term continuous rainfall data from meteorological stations within the study area. In addition, this study adopts the approach of [89] by using the mean annual rainfall from monthly data to reduce the rainfall distribution variation.

- *Soil Erodibility (K) factor*

One of the main issues in estimating soil erosion is the lack of data on soil features. K-factor (MJ-1 mm$^{-1}$), as used in the soil erosion model, the Universal Soil Loss Equation (USLE) and its revised version (RUSLE), can be used as a key factor to model soil, based on soil erodibility. More particularly, this factor, which expresses the susceptibility of a soil to erosion, is related to soil properties such as organic matter content, soil texture, soil structure and permeability. The K factor calculation based on LUCAS datasets is provided from JRC [52].

- *Topographic Slope Length & Steepness (LS) factor*

The LS factor indicates the effect of topography on soil erosion and describes the slope length (L) and the steepness (S) parameters. The former (L) is defined as the point of departure of the surface runoff to the point, while the latter (S) describes the behavior of soil erosion with an inclination angle [83].

Ref. [90] developed an approach that combines both parameters to estimate the LS factor. In this way, input data is distinguished in the upslope contributing area per unit width, determined by the flow accumulation, the pixel size, and the slope, while the outcome is unitless (Equation (6)).

$$LS = \left(\frac{U}{L0}\right)^m \times \left\{\left[\sin\left(\frac{\beta \times 0.01745}{S0}\right)\right]^n\right\} \times (m+1) \tag{6}$$

where *U* is the flow accumulation multiplied by the pixel size, *L0* is the slope length (22.13 m), *β* is the slope in degrees, *S0* is the slope percentage (9%), *m* is sheet erosion ranging from 0.4 to 0.6, accordingly to the slope intensity, and *n* is rill erosion ranging from 1 to 1.3. The values of the indicators used are *m* = 0.5 and *n* = 1.2 [87].

- *Cropping and Land-Cover (C) factor*

The C factor describes the influence of soil cover, crop and management territorial loss in relation to the territorial loss in bare fallow land areas [86]. It is a dimensionless factor ranging from 0, in case of intense coverage, to 1 when there is significant lack of coverage in vegetation.

Due to the important relationship between soil and vegetation coverage, Durigon et al. [91] provided the following Equation (7) to describe the *C* factor based on the Normalized Difference Vegetation Index (NDVI):

$$C = \frac{1 - NDVI}{2} \tag{7}$$

- *Conservation Practices (P) factor*

The support Practice (P) factor is defined as the ratio of soil loss in a specific soil conservation practice to a rough field. Thus, factor P is important to take into consideration, as it can provide information about what practices are more advantageous for soil conservation [84].

Concerning the values of the P factor, these range from 0.2 to 1, with low values corresponding to greater control of soil erosion. P values can be extracted from the literature, while in other cases P is empirically considered. For example, a value of 1 for the P factor means that either there are no support practices or there are conventional techniques. On the other hand, a value of 0.25 shows the potential for this management factor to reduce soil by a 75% loss [84]. P factor can also be estimated based on the Corine Land Cover (CLC) data. According to Yang et al. [92] for Corine land categories (211, 212, 221–223, 231, 241–243), P is estimated as 0.5, which according to David [93] corresponds to "minimum plowing", while all other classes were given the value of 1.

### 3.6. Tsunami Run-Up Scenario

Tsunami events are extremely destructive natural hazards, which may occur rarely, but cause extreme loss of human life, as well as of structural properties. This makes the estimation of the parts of a coastal zone more susceptible to tsunami events a necessity when studying the hazard susceptibility of an area. It is worth mentioning that tsunamis can be generated from underwater earthquakes, volcanic eruptions, landslides, or extraterrestrial impacts such as asteroids. In our work, the tsunami run-up model was based on seismogenic tsunamis in an attempt to visualize the worse-case-scenario, as suggested by the research of Batzakis et al. [38]. This model was created using tectonic plate seismic data based on the earthquake magnitude most associated with high tsunami run-up. This specific threshold was suggested by Iida (1963) as approximately Mw~8.5 [94]. These magnitude values were associated with a wave of 10m mean maximum run-up, which has been confirmed by several studies in the coastal zones of the Cyclades islands [38]. Through the application of a GIS, the elevation pixel values of the utilized DEM were transformed into inundation depths, divided into five categories using natural breaks (Jenks).

### 3.7. Analytical Hierarchy Process

The total susceptibility of coastal regions to natural hazards was estimated by using the Analytical Hierarchy Process (AHP) method. The AHP is a decision-making analytical methodology, which excels in solving multi-criteria problems [95,96]. In our work. the four hazards represent the problem variables.

As a second stage, it was necessary to compare these variables in pairs (pair-wise comparison) based on their impact on the overall problem. This pair-wise comparison approach places the separate hazards in a hierarchy, via a comparison matrix [96]. Once the comparison matrix is completed, the weights of importance for each hazard value were calculated, which represent their influence. This was performed by dividing the geometric mean of each line ($u_i$) by the sum of the geometric means of all rows ($u_k$) of the matrix, via the following Equation (8):

$$w_i = u_i / \sum_{k=1}^{n} u_k, \tag{8}$$

The consistency of the pairwise comparisons of the square matrix can be evaluated via the Consistency Index (CI) and the Consistency Ratio (CR). The CI was calculated via Equation (9):

$$CI = \frac{(\lambda_{max} - n)}{(n-1)}, \tag{9}$$

where $\lambda_{max}$ is the largest eigenvalue of the matrix and $n$ is the order of the matrix.

Meanwhile, the CR is calculated through the following Equation (10):

$$CR = \frac{CI}{RI}, \tag{10}$$

where the *RI* is the randomness index value that depends on the order of the matrix published by Saaty [95], and *CI* is the Consistency Index. The CR needs to be lower than 0.1 and realistically higher than 0 in order for the comparison to be considered consistent, and thus successful. In this work, CR value was calculated with an acceptable consistency at 0.01. The calculated weights were used as multiplying factors for each classified criterion, in order to create the final susceptibility map.

## 4. Results

In this study, four criteria corresponding to the most common hazards on both islands were selected for homogenization and incorporation into the presented AHP implementation.

### 4.1. Susceptibility to Landslides

Applying the RES to determine the susceptibility of Milos and Thira islands to landslides, a 10 × 10 interaction matrix (Table 4) was implemented. It is important to note that the same comparison ranks were used for the causal factors of both islands due to their similarities, regarding their interaction intensity (C+E). The factors were categorized as P1 lithology, P2 mean annual precipitation, P3 slope gradient, P4 solar radiation, P5 earthquake density, P6 earthquake depth, P7 distance from the road network, P8 proximity to tectonic structures, P9 proximity to the drainage network and P10 slope curvature.

**Table 4.** RES interaction matrix.

| P1 | 0 | 1 | 0 | 0 | 2 | 0 | 4 | 3 | 3 |
|----|----|----|----|----|----|----|----|----|-----|
| 4 | P2 | 2 | 4 | 0 | 0 | 2 | 2 | 4 | 4 |
| 4 | 0 | P3 | 1 | 0 | 1 | 3 | 4 | 3 | 0 |
| 4 | 2 | 1 | P4 | 0 | 0 | 0 | 0 | 0 | 0 |
| 4 | 0 | 1 | 0 | P5 | 4 | 0 | 3 | 0 | 3 |
| 3 | 0 | 2 | 0 | 4 | P6 | 0 | 3 | 0 | 2 |
| 0 | 0 | 4 | 0 | 0 | 0 | P7 | 0 | 0 | 3 |
| 0 | 0 | 3 | 1 | 3 | 3 | 0 | P8 | 0 | 3 |
| 2 | 0 | 2 | 0 | 0 | 0 | 2 | 0 | P9 | 2 |
| 2 | 0 | 1 | 3 | 0 | 0 | 0 | 0 | 0 | P10 |

After determining the interactions between the 10 causal factors, a sample of random landslides equal to 70% of the landslide inventory was used to calculate the weights of importance for each factor. The landslide inventory was mainly obtained using optical observations in Google Earth Pro and bibliographic data [63,97]. A total of 162 landslides were observed in Milos and Thira, respectively, within the boundaries of the low-lying coastal areas.

The remaining 30% were used to validate the results of the landslide susceptibility model. The C+E% values were divided by the maximum susceptibility value (Max Pij) of each factor in order to calculate the necessary weights (Wi) (Table 5).

Once the weights of importance were calculated, the factor layers were multiplied by their respective weights and then added together to generate the landslide susceptibility models.

Based on the RES analysis and processing of the existing landslides inventory in GIS, the landslide susceptibility maps of Milos and Thira Islands (Figure 6) were generated as illustrated in the following figures.

Due to the similar morphology of both islands, which is characterized by relatively steep terrain slopes, landslides are the most common threat. However, only a small percentage of the entire low-lying coastal zones fall within the high and very high susceptibility classes. In particular, on the island of Milos, the high (13.79%) and extremely high (6.25%) susceptibility classes cover 19%, while on the island of Thira they cover only 13% (Table 6).

**Table 5.** Interactive intensity, dominance and weighted coefficients of the principal factors in the RES method for Milos and Thira Islands.

| Landslide Factors | Interactive Intensity C+E | Dominance C − E | C+E % | Max. Pij Rating | Wi |
|---|---|---|---|---|---|
| P1 | 36 | −10 | 14.88 | 5 | 0.595 |
| P2 | 24 | 20 | 9.92 | 5 | 0.397 |
| P3 | 33 | −1 | 13.64 | 5 | 0.546 |
| P4 | 16 | −2 | 6.61 | 5 | 0.264 |
| P5 | 22 | 8 | 9.09 | 5 | 0.364 |
| P6 | 24 | 4 | 9.92 | 5 | 0.397 |
| P7 | 14 | 0 | 5.79 | 5 | 0.232 |
| P8 | 29 | −3 | 11.98 | 5 | 0.479 |
| P9 | 18 | −2 | 7.44 | 5 | 0.298 |
| P10 | 26 | −14 | 10.74 | 4 | 0.671 |

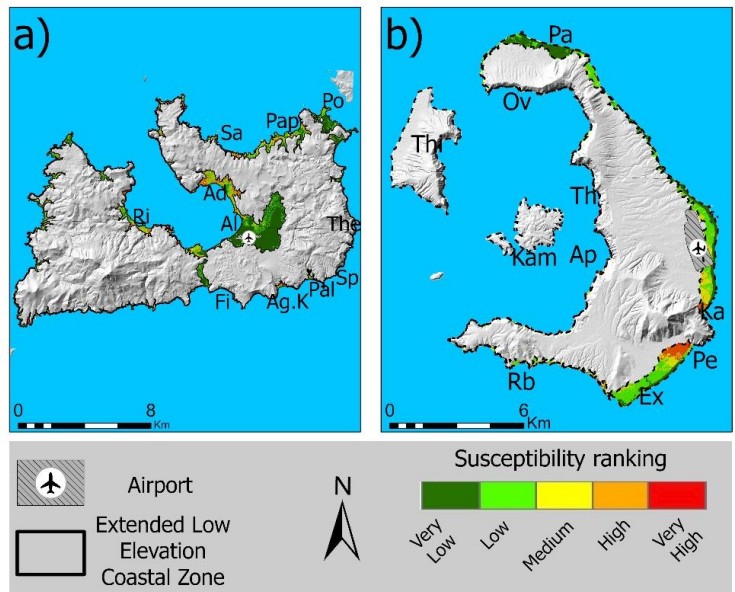

**Figure 6.** Landslide susceptibility maps for Milos (**a**) and Thira (**b**) islands.

**Table 6.** Percentages of susceptibility classes to landslides in Milos and Thira islands.

| | Very Low | Low | Medium | High | Very High |
|---|---|---|---|---|---|
| Milos | 34.87 | 24.09 | 20.98 | 13.79 | 6.25 |
| Thira | 28.19 | 35.90 | 22.56 | 11.14 | 2.20 |

Specifically, the beaches of Agia Kyriaki (Ag. K), Filotas (Fi), and Palaiochori (Pal) on the southeast side of the island of Milos were the most vulnerable. Concerning the island of Thira, areas of high threat were identified in the southern parts of the coastal region in the Perissa (Pe) and Kamari (Ka) settlements, as well as Red beach (Rb) (Figures 7 and 8).

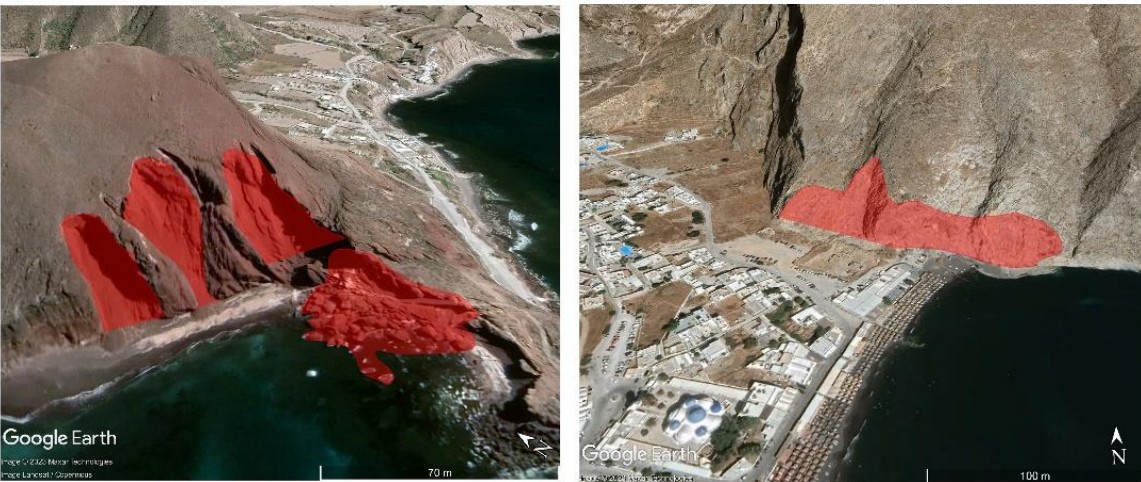

**Figure 7.** Google earth images representing two coastal areas on eastern Thira near to active or potential landslide triggering. (Red Beach on the **left** and Perissa on the **right**).

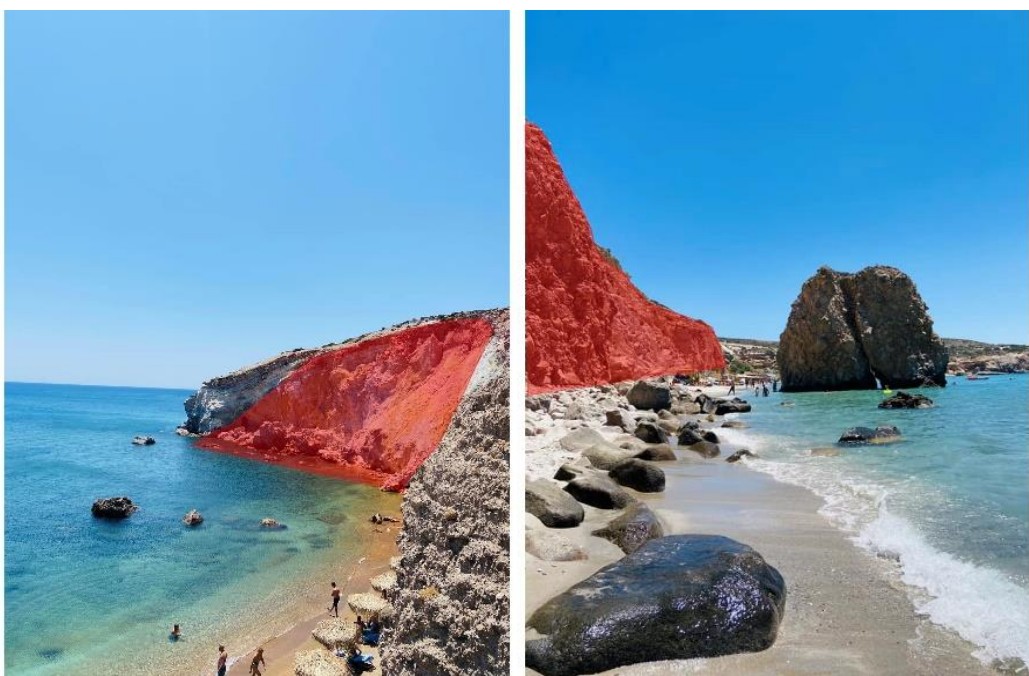

**Figure 8.** Images representing two coastal areas on eastern Milos nearby to active or potential landslide triggering. (Firiplaka on the **left** and Palaiochori on the **right**).

*4.2. Susceptibility to Torrential Floods*

Once all the conditioning factors were processed into a common scale, the four thematic layers were combined through map algebra, using the following Equation (11):

$$\text{FFPI} = \frac{(\text{Slope} + \text{Landcover} + \text{Soiltype} + \text{Vegetationdensity})}{4} \tag{11}$$

Due to their significant influence on the coastal area, the causal factors studied were considered equal when calculating the FFPI values of the islands. The resultant values were on a scale of 1 to 5, representing the increasing flash flood potential and overall torrential flood susceptibility of the study areas (Figure 9).

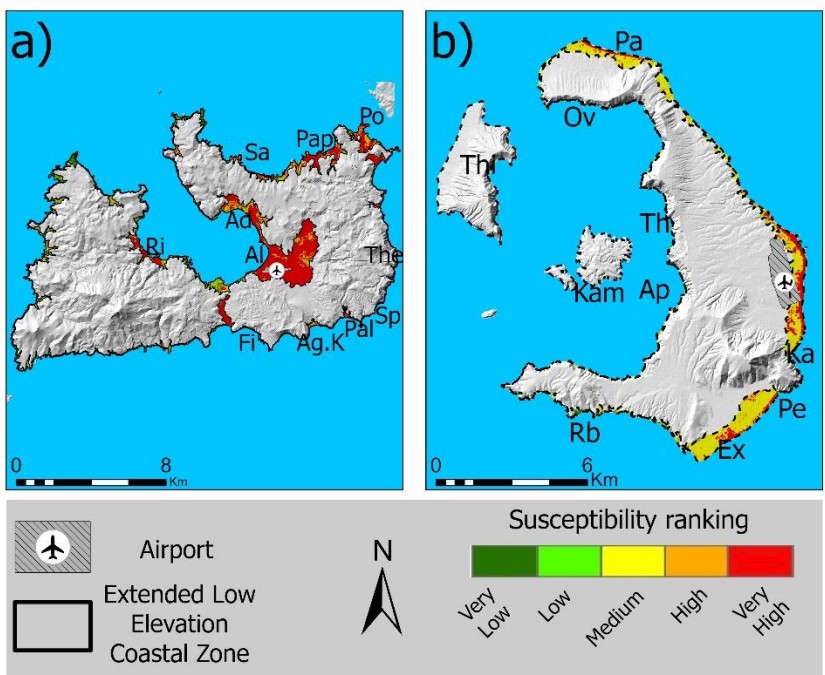

**Figure 9.** Torrential flood susceptibility maps in Milos (**a**) and Thira (**b**) islands.

A large part of the coastal region of Milos Island has a high (24.04%) and very high (50%) susceptibility to the occurrence of torrential floods (Table 7). In the north and west of the island, very low, low, and moderate susceptibility are predominant. In terms of the onshore regions of Thira Island, the eastern and southern regions have high (10.78%) and extremely high (19.50%) susceptibility values. The medium susceptibility class has the highest coastal coverage at 59.89%, followed by the very low and low classes at 5.11% and 4.71%, respectively.

**Table 7.** Percentages of susceptibility classes to torrential flood in Milos and Thira islands.

|       | **Very Low** | **Low** | **Medium** | **High** | **Very High** |
|-------|--------------|---------|------------|----------|---------------|
| Milos | 16.81        | 4.68    | 4.44       | 24.04    | 50            |
| Thira | 5.11         | 4.77    | 59.82      | 10.78    | 19.50         |

*4.3. Soil Loss*

The soil erosion model was created by using the RUSLE equation to combine the soil loss causal factors. After generating the required layers, the result was a soil loss map of Milos and Thira's coastal zones (Figure 10).

The erosion susceptibility maps (Figure 10) depict low potentiality in both islands. Although, most of the coastal areas are characterized by low erosion that ranges between 94.94–95.28% (Table 8), the high and very high susceptibility values were identified near to the drainage network [89].

*4.4. Susceptibility to Tsunami*

As stated by Batzakis et al. [38], the inundation depth of the zones required to be estimated in order to develop the tsunami susceptibility model. The results were limited to a maximum elevation of 10 m, because this is considered the maximum possible tsunami run-up, and classified into five categories. The susceptibility maps (Figure 11) illustrate the coastal regions that are more vulnerable to tsunami. The high and very high classes, in particular, cover up to 20% of Milos Island and approximately 26% of Thira Island (Table 9). Specifically, in Milos Island, the high susceptibility classes cover the central onshore areas

at Alikes (Al), Adamas (Ad) settlement and Rivari Lagoon (Ri). as well as on the northern side in Papafragas (Pap) and Polonia (Po) settlements. In Thira Island, the most susceptible areas are indicated on the southeastern sides in Exomitis (Ex), Perissa (Pe) and Kamari (Ka) settlements.

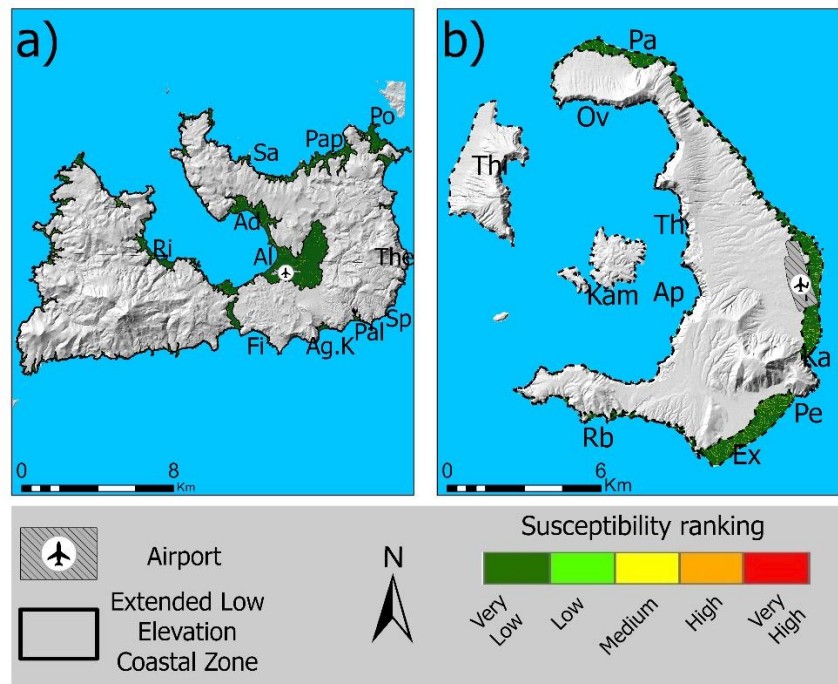

**Figure 10.** Erosion susceptibility maps in Milos (**a**) and Thira (**b**) islands.

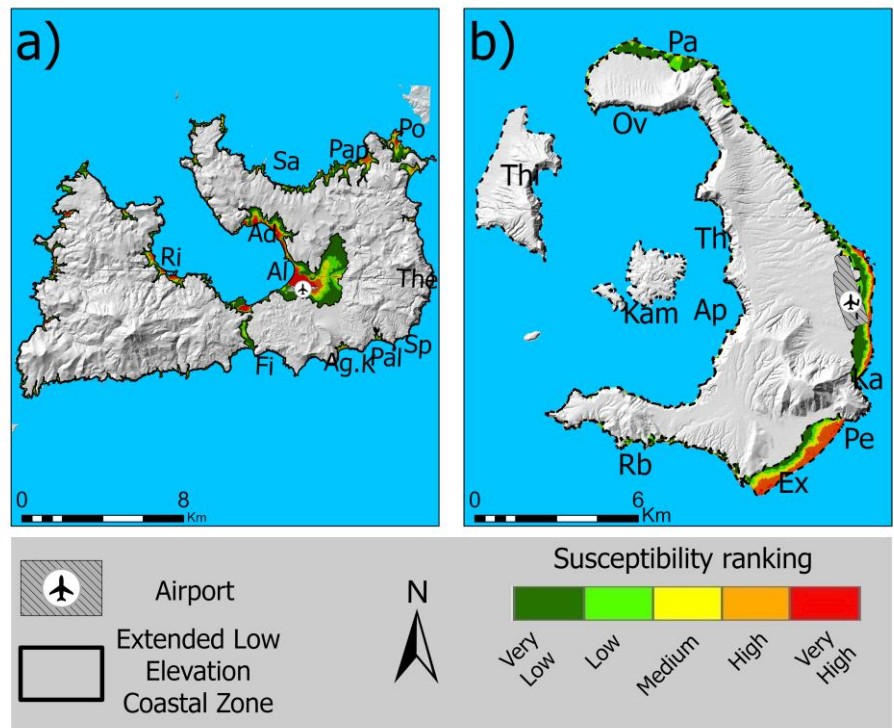

**Figure 11.** Tsunami susceptibility maps in Milos (**a**) and Thira (**b**) islands.

**Table 8.** Percentages of susceptibility classes to soil erosion in Milos and Thira islands.

|       | Very Low | Low  | Medium | High | Very High |
|-------|----------|------|--------|------|-----------|
| Milos | 95.28    | 3.12 | 0.97   | 0.46 | 0.15      |
| Thira | 94.94    | 3.78 | 0.85   | 0.28 | 0.12      |

**Table 9.** Percentages of susceptibility classes to tsunami run-up scenario in Milos and Thira islands.

|       | Very Low | Low   | Medium | High  | Very High |
|-------|----------|-------|--------|-------|-----------|
| Milos | 43.69    | 20.25 | 15.25  | 10.71 | 10.08     |
| Thira | 36.70    | 19.73 | 16.87  | 20.31 | 6.36      |

*4.5. Total Susceptibility to Hazards*

The AHP methodology was utilized to determine the overall susceptibility to multiple natural hazards using a 4 × 4 comparison matrix (Table 10), in order to calculate the weights of importance for each criterion (hazard) using Equation (12). According to the applied scenario, landslides and torrential floods are the most prevalent threats, with landslides being the most prevalent. This is because both islands are frequently affected by landslides, as evidenced by the large number of past landslide events in the two islands, as well as by the numerous debris and mud flows mapped on the geological maps. In addition, although less frequent than landslides, flood events have been recorded, especially in Thira in recent years, making this a critical hazard. As the coastal environment is subject to a constant barrage of erosion processes, soil loss is a greater threat than severe seismogenic tsunamis, due to their high return period. According to [98], the return period values for shallow earthquakes of magnitude M ≥ 6.0 in the South Aegean is more than 200 and/or 400 years.

**Table 10.** Assigned values and comparison between all hazards. Ui depicts the geometric mean of normalized values of each line.

|              | Landslide | Flood | Soil Erosion | Tsunami | $U_i$  | Weights |
|--------------|-----------|-------|--------------|---------|--------|---------|
| Landslide    | 1.00      | 2.00  | 3.00         | 4.00    | 0.4637 | 0.4669  |
| Flood        | 0.50      | 1.00  | 2.00         | 3.00    | 0.2757 | 0.2776  |
| Soil erosion | 0.33      | 0.50  | 1.00         | 2.00    | 0.1592 | 0.1603  |
| Tsunami      | 0.25      | 0.33  | 0.50         | 1.00    | 0.0947 | 0.0953  |

Particularly, landslides were assigned with the greatest weight of importance value (0.4669). Intense precipitation is rare in the Cyclades, but in the context of climate change, the islands are prone to intensive torrential flood events that can cause damage, especially in low-lying areas. This hazard was assigned a weight value of 0.2776. Moreover, soil erosion was assigned with 0.1603 and tsunami with 0.0953. Based on the obtained weight values (Equation (12)), total susceptibility to the assessed hazards was calculated, utilizing the following Equation:

$$AHP = 0.4668 \times L + 0.2776 \times F + 0.1603 \times S + 0.0953 \times T \tag{12}$$

where *L* represents landslides, *F* torrential floods, *S* soil erosion and *T* tsunami. The results (Figures 12 and 13) were reclassified into five classes using the natural breaks (Jenks) method. Green color illustrates the areas with the lower susceptibility score and red color the higher, respectively.

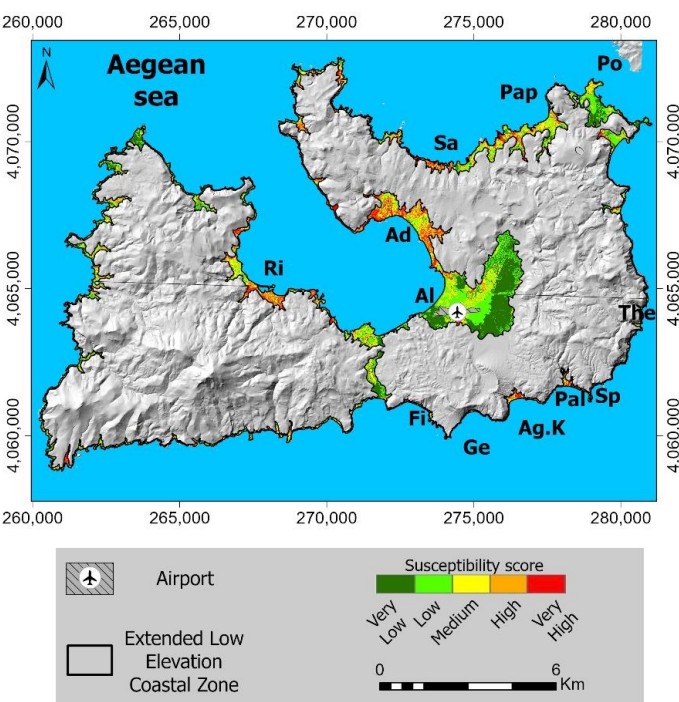

**Figure 12.** AHP ranking of Milos island's extended low elevation coastal zone multi-hazard susceptibility map.

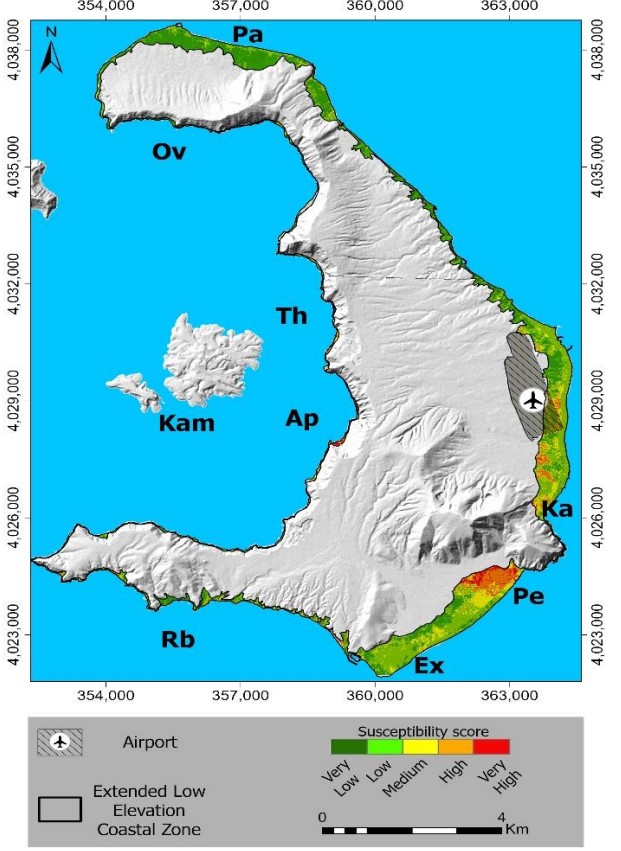

**Figure 13.** AHP ranking of Thira island's extended low elevation coastal zone multi-hazard susceptibility map.

The results indicated that the highly scored areas were located in the Adamas (Ad) settlement, Sarakiniko (Sa) beach and the Rivari Lagoon (Ri) (Figure 14).

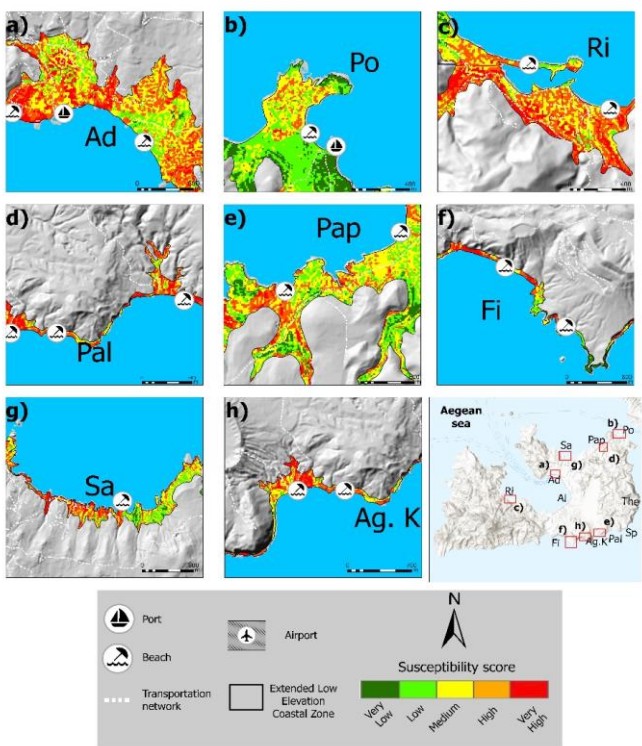

**Figure 14.** Map illustrating areas (**a–h**) prone to AHP multi-hazard in Milos Island. Ad: Adamas, Po: Posidonia, Ri: Rivari lagoon, Pal: Palaiochori, Pap: Papafragas, Fi: Firiplaka, Sa: Sarakiniko and Ag.K: Agia Kiraki. Green color illustrates the areas with the lower susceptibility score and red color the higher, respectively.

Particularly, the analysis of the results for Milos illustrated (Table 11) that 25.85% (0.24 km$^2$) of the built-up areas, 56.46% (0.05 km$^2$) coverage of the ports, 3.77% of the airport, 35.48% (0.35 km$^2$) of beaches and 20.44% (0.05 km$^2$) of the main transportation network are characterized by high susceptibility (Figure 14).

**Table 11.** Multi-hazard susceptibility statistics in Milos and Thira islands regarding the natural and manmade environments.

| | **High & Very High Susceptibility** | |
| --- | --- | --- |
| **Land Cover** | **Milos (%)** | **Thira (%)** |
| Built-up area | 25.85 | 33.38 |
| Port area | 56.46 | 44.51 |
| Airport area | 3.77 | 33.02 |
| Beach | 35.48 | 12.76 |
| Transportation network | 20.44 | 22.19 |

Regarding the results for Thira Island, the highly scored areas were identified on the west side, where geomorphology is characterized by high relief slope. Furthermore, high values were also detected in the Perissa (Pe) settlement, which is located on the northeast side of Thira (Figure 15). Specifically, 33.38% (0.36 km$^2$) of the built-up areas, 44.51% (0.03 km$^2$) of ports, 33.02% (0.09 km$^2$) of the airport, 12.76% (0.04 km$^2$) of beaches and 22.19% (0.04 km$^2$) of the main transportation network are characterized by high susceptibility.

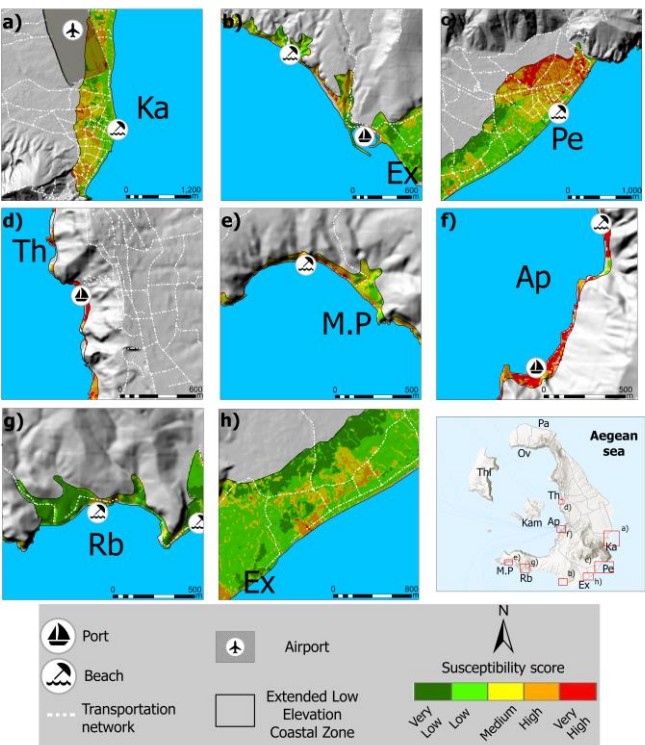

**Figure 15.** Map illustrating areas (**a**–**h**) prone to AHP multi-hazard results in Thira Island. Ka: Kamari, Pe: Perissa, Ap: Athinios, M.P: Mesa Pigadia beach, Rb: Red beach. Green color illustrates the areas with the lower susceptibility score and red color the higher, respectively.

*4.6. Result Validation*

The validation of the total susceptibility is a complex problem; therefore, the models' validation was accomplished by validating each hazard individually. Based on 30% of the landslide inventory, the percentage success of RES methodology was 97.37%.

In the case of susceptibility to torrential floods, areas with high and very high zones were confirmed from recent flood events near to the airport area and Kamari settlement. There was no data for validation on Milos. However, recent news from online local newspapers have announced the establishment of new anti-flooding works in Milos for the upcoming period.

Despite the fact that the soil loss values are low, the higher values were identified in specific areas and especially along the existing drainage network, following the pattern of the LS factor. Particularly, high erosion values were detected between 2016 and 2021 in Kamari (Figure 16c,d) settlement (eastern Thira) and adjacent to the southeastern part of the airport (Figure 16a,b).

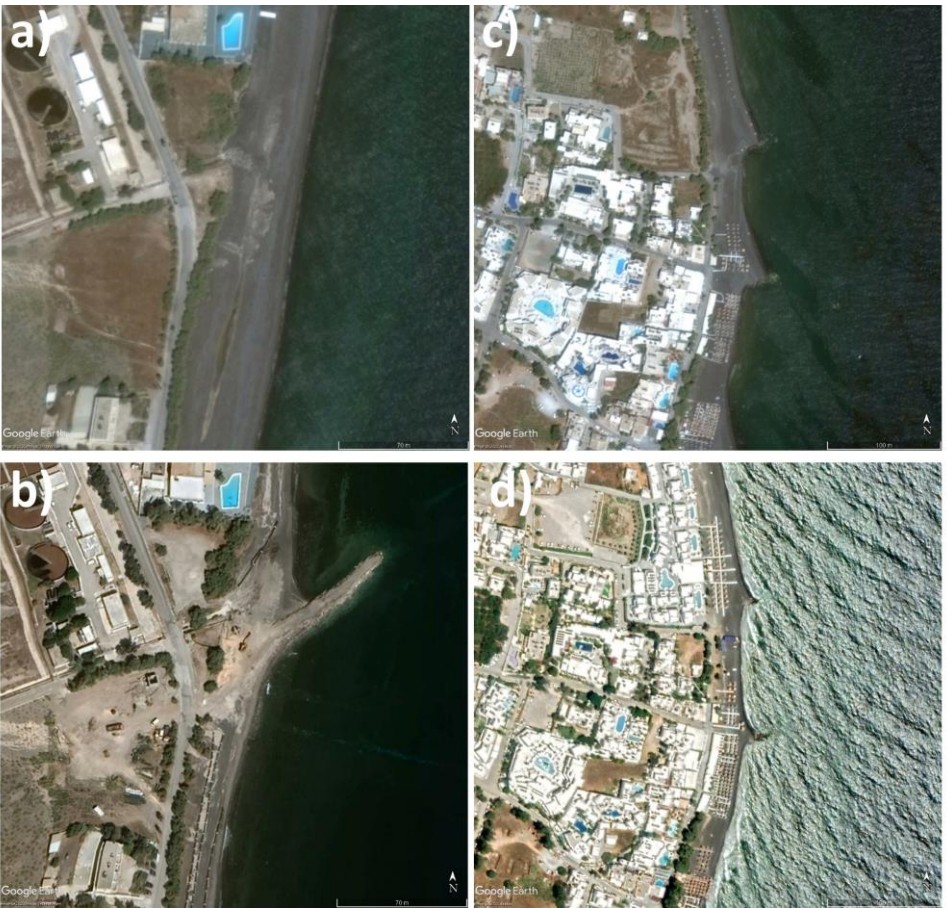

**Figure 16.** Satellite images between 2016 and 2021 based on Google Earth pro platform. Figures (**a**,**b**) correspond to the airport area, while (**c**,**d**) depict the high urbanization rate in the Kamari settlement.

Finally, the tsunami susceptibility model depicted that low-lying coastal areas on both islands are vulnerable to tsunami, especially the areas adjacent to the coastline and with lower elevation values. Specifically, Kamari, Perissa and the eastern part of Thira are more susceptible to tsunami hazard. It is worth mentioning that Kamari settlement was developed after the 1956 earthquake, due to the relocation of the population from other settlements of the island that were destroyed by this very strong earthquake, following the initial Hellenic Anti-Seismic Regulation in 1959 [99]. Moreover, terrain vulnerability to tsunami in Milos is higher in Adamas (central port), Alikes (central Milos) and the southeastern part, where high tourism activity and marine activities are concentrated annually.

## 5. Discussion and Conclusions

The SAVA region is one of Europe's most seismic and volcanic landscapes, making it a one-of-a-kind physical laboratory for researchers and visitors. Additionally, reports have shown that the Aegean Sea will be one of the most vulnerable areas in the Mediterranean basin to global warming and climate change scenarios, with more extreme meteorological events, heat waves and droughts. Apart from that, the Cyclades complex and the South Aegean region was the only area in Greece with a population rise of more than 5% from 2011 to 2021. Thus, the study areas are a hub of human-caused activity, especially during the tourist season, which starts from April and ends in October, annually.

This work introduces a workflow to determine the most susceptible regions, in terms of multi-hazard susceptibility, for low-lying coastal areas in Thira and Milos islands. A multi-hazard methodology is a complex work, due to the fact that single hazards with different approaches must be compared in a specific period and specific geographic boundaries. Our

pair-wise comparison and applied methodologies for terrain susceptibility to landslide, torrential flood, soil erosion, and tsunami yielded the following results.

Initially, the landslide susceptibility map was derived by implementing and integrating the RES approach. Especially, for this hazard a landslide inventory was created in order to validate the model on both volcanic islands. The calculated results illustrated that more than 55% of the terrain was found within the very low and low susceptibility zones, and more than 10% was found within the high and very high susceptibility zones (Table 6). In particularly, the visualized outcomes highlighted those southeastern areas of Milos, such as Ag. Kiriaki, Filotas, Palaiochori and the western part of Adamas settlement, as the most landslide-affected areas (Figure 6). Additionally, the areas most prone to landslides are located in southeastern regions of Thira, such as Perissa, Kamari, Red beach, Athinos port (central region), and Oia's desalination plant (northwest side). These findings were driven due to the interaction and weighting analyses of 10 parameters, such as seismicity, lithological conditions, terrain curvature, soil radiation, etc.

According to Figure 9, the high and very high flood susceptibility zones are located in low-lying regions. The results illustrated that more than 10% of the terrain was found within the very low and low susceptibility zones, and more than 70% (Milos) and 30% (Thira) were found within the high and very high susceptibility zones (Table 7). The northern areas of Milos, such as Sarakiniko, Papafragas and Pollonia settlements, central regions, e.g., Adamas and Alikes, and the western area of Firiplaka settlement are occupied by agricultural activities (Figure 9). Additionally, the most flood-affected areas are located in southeastern regions of Thira such as Perissa, Kamari and Exomitis settlements (Figure 9).

The calculated values were quite low on both islands in terms of soil loss assessment. Specifically, more than 90% was mapped within the low and very low susceptibility zones. In contrast to these results, regions with high soil loss values were identified in the vicinity of the existing drainage network following an LS pattern. According to Figure 10, prone areas were identified in southeastern regions such as Agia Kiriaki, Paliochori, Spathi and Thiorichio. Moreover, in Thira, eastern and southeastern areas were found to be more susceptible to flood, such as Exomitis, Perissa, Kamari, and areas near to the southeast side of the airport.

The tsunami susceptibility map was generated based on the conversion of low elevation values up to 10 m. Coastal regions of the Cyclades islands complex have been affected historically by a run-up of 10 m. According to the worse-case run-up scenario of 10, the results illustrated that more than 55% of the terrain was found within the very low and low susceptibility zones in both case study areas (Table 9). In particular, the visualized outcomes highlighted the central area of Milos, such as Adamas and Alikes, and the northeastern side, such as Pollonia settlement (Figure 11). Additionally, the tsunami-affected areas are located in eastern and southeastern regions of Thira, such as Perissa, Kamari and Exomitis, and adjacent to the airport facility (Figure 11).

According to Figures 12 and 13, two multi-hazard susceptibility maps were produced within the low-lying coastal limits. Although the percentage of the most vulnerable regions on both islands is less than 25%, the majority of ports and transportation networks are located in the high and very high susceptibility zones. According to Table 11 and the adopted scenario, high and very high zones include more than 30% of the built-up area, 20% of the transportation network, and 50% of the seaports. Notable is the fact that 33% of Thira's airport and more than 30% of Milos's beaches are vulnerable to severe hazards. Further to that, the aforementioned findings highlight the necessity for scenarios and potential threats modelling, during a potential hazard event in low-lying areas.

This work has some limitations due to the fact that it had to compare four specific hazards with selected criteria. Furthermore, the ranking of the criteria in AHP analysis was based on the appearance frequency of the hazards. Finally, this study presents only one adopted scenario, considering landslide susceptibility as the most important hazard.

Summarizing, this study highlights the importance of using earth observation and geo-information methods to investigate hazard assessment in low-lying coastal areas,

on a regional scale. Future research could be based on multi-hazard integration with other hazards, such as volcanic activity and hydrothermal investigation in southern Milos, where the EU Pathfinder project RAMONES will research the offshore area in 2023 using innovative prototype instruments. Therefore, advanced remote sensing methods in terms of ground subsidence (e.g., InSAR methodology) for shoreline changes utilizing the Digital Shoreline Analysis System (DSAS) will be a part of future work, in order to investigate potential correlation. The identification of more vulnerable areas to hazards using multi-hazard approaches could be used as a baseline for civil protection, regional planning and decision making. Thus, the results of this work could be combined with data from innovative technological instruments such as High-frequency (HF) radar that monitors the sea surface velocity and provides information for tsunami early warning and SLR [100].

**Author Contributions:** Conceptualization, P.K., A.K. and I.P.; methodology, P.K., A.K. and I.A., validation, P.K., N.K., P.N., I.A. and K.K.; formal analysis, P.K., K.K. and N.K.; investigation, P.K., A.K., K.K. and N.K.; resources, P.K. and N.K.; data curation, P.K., A.K., K.K. and P.N.; writing—original draft preparation, P.K.; writing—review and editing P.K., A.K., P.N., K.K., N.K., I.P., I.A. and S.K.; visualization, P.K. and A.K.; supervision P.K. and N.K. All authors have read and agreed to the published version of the manuscript.

**Funding:** This research received no external funding.

**Data Availability Statement:** Data available upon request.

**Acknowledgments:** Maps and diagrams throughout this work were created using ArcGIS® software by Esri. ArcGIS® and ArcMap™ are the intellectual property of Esri and are used herein under license. Copyright © Esri. All rights reserved. For more information about Esri® software, please visit https://www.esri.com/en-us/home, accessed on 10 March 2023. Many thanks are also given to the colleague Anezina Solomonidou, specializing in planetary geology for her contribution regarding photos and information for Milos Island. The authors are grateful to the European Space Agency who provided Sentinel-2 products. The authors would also like to thank the reviewers for providing useful suggestions that enhance the manuscript's quality.

**Conflicts of Interest:** The authors declare no conflict of interest.

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
