# Peer review of "Multi-Hazard Susceptibility Assessment Using the Analytical Hierarchy Process in Coastal Regions of South Aegean Volcanic Arc Islands"

_2624-795X, doi:10.3390/geohazards4010006_

Round 1

Reviewer 1 Report

The study analyzed the onshore coastal terrain’s susceptibility to natural hazards and identified regions that are vulnerable to soil erosion, torrential flooding, landslides, and tsunamis. An originality of this study is that the Multi-Criteria Decision-Making (MCDM) analysis was adopted for susceptibility assessment. I agree that this work is important to early warning systems in the Aegean Sea area and can be extended to other regions. It can be a great contribution to literature. In general, the manuscript was logically organized, and the results supported the conclusion. Therefore, my recommendation is minor revision.

My main comment is about the conditioning factors of landslide. I agree that lithology, slope gradient, or distance from tectonic structure are important factors. However, I cannot agree that distance from the main road network is regarded as a casual factor (Table 2). From my point of view, this parameter has nothing to do with the occurrence of landslide, although the landslide may affect the transportation. Please consider removing this causal factor from the analysis, or please offer convincing explanations on its reason.

Other minor comments are as follows.

(1)  Line 77: Previous research has been conducted on tsunami hazards in SAVA area. You may refer to the following publication in this paragraph.

https://doi.org/10.5194/nhess-7-57-2007

(2)  Line 97: The sentence seems confusing. “this work introduces a multi-hazard approach that could act as a baseline for multi-hazard approaches…” You may change its expression to avoid repeating.

(3)  Figures 2 and 3: Please use longitude and latitude to show the range of study region. Please specify the meaning of appreciations in the figure (e.g., Pap, Sa).

(4)  Line 400: Besides tectonic earthquakes, landslide or volcanic eruption can also generate tsunamis resulting in inundation. Although the model adopted in this study only considers seismogenic tsunami, other types of tsunamis should be mentioned in the text.

(5)  Line 446: This matrix is not a symmetric matrix. How did you distinguish the relationship (cause or effect) between two factors? Please specify.

(6)  Line 548: The determination of weight values (Equation 13) was not explained. More details are expected on how to determine the weight of each hazard.

(7)  Line 610: This work is important to early warning systems in the wider are of the Aegean Sea, but in this article, it lacked detailed explanations on the potential early warning systems. Discussing the potential early warning system can highlight the novelty and contribution to the manuscript. Here I recommend two papers on tsunami early warning system that can be referred.

https://doi.org/10.1029/2022JB025153

I look forward to the acceptance and publication of this work!

Author Response

Dear reviewer first of all thank you four useful and insightful comments. I want to thank you also for the opportunity to submit the revised version of the paper " Multi-hazard susceptibility assessment using the Analytical Hierarchy Process in Coastal Regions of South Aegean Volcanic Arc Islands" in GeoHazards Journal. We are grateful about the time and effort that you dedicated to providing feedback on our manuscript. We are also glad about the helpful comments and the suggestions made to improve our manuscript. Please see the following section for a detailed response to the reviewers' comments and concerns.

My main comment is about the conditioning factors of landslide. I agree that lithology, slope gradient, or distance from tectonic structure are important factors. However, I cannot agree that distance from the main road network is regarded as a casual factor (Table 2). From my point of view, this parameter has nothing to do with the occurrence of landslide, although the landslide may affect the transportation. Please consider removing this causal factor from the analysis, or please offer convincing explanations on its reason.

Author’s response: Thank you very much for your very useful notice. All the factors that were used for the landslide susceptibility methodology, were implemented taking into consideration the total extent on the two islands. There are areas near low-lying regions (such as ports, beaches, etc) where the existing road network can cause instabilities to the adjacent slopes. Thus, proximity to road network named as P7 in Table 5, has the minimum weight in comparison with the other factors and has the lowest importance. In other environments, such as in high altitudes (mountainous regions) with very high precipitation heights this factor will have a higher rank or weight regarding the total landslide susceptibility.

Other minor comments are as follows.

(1)  Line 77: Previous research has been conducted on tsunami hazards in SAVA area. You may refer to the following publication in this paragraph.

https://doi.org/10.5194/nhess-7-57-2007

Author’s response: Dear reviewer, thank you for your suggestion! We included this reference to our manuscript.

(2)  Line 97: The sentence seems confusing. “this work introduces a multi-hazard approach that could act as a baseline for multi-hazard approaches…” You may change its expression to avoid repeating.

Author’s response: Dear reviewer, thank you for pointing this out. We revised it accordingly.

(3)  Figures 2 and 3: Please use longitude and latitude to show the range of study region. Please specify the meaning of appreciations in the figure (e.g., Pap, Sa).

Author’s response: Thank for highlighting this! Indeed, we should specify the abbreviations that presented in Figure 2 and 3. We made the necessary modifications.

(4)  Line 400: Besides tectonic earthquakes, landslide or volcanic eruption can also generate tsunamis resulting in inundation. Although the model adopted in this study only considers seismogenic tsunami, other types of tsunamis should be mentioned in the text.

Author’s response: Dear reviewer, indeed this study considers only the seismogenic tsunami in the scenario. We have mentioned the other types of tsunamis that can be triggered by another factors such submarine landslides or volcanic activity.

(5)  Line 446: This matrix is not a symmetric matrix. How did you distinguish the relationship (cause or effect) between two factors? Please specify.

Author’s response: Dear reviewer, thank you for your comment, we have revised the matrix accordingly. The scoring between two factors has been based on expert’s opinion, and literature. For example, the comparison between P1 (lithology) and P2 (precipitations) has been implemented scoring the interaction (counterwords) from lithology to precipitation as zero (no interaction) and from precipitation to lithology as four (strong interaction).

(6)  Line 548: The determination of weight values (Equation 13) was not explained. More details are expected on how to determine the weight of each hazard.

Author’s response: Dear reviewer, the determination of weight values is explained in subsection 3.7. Particularly, in Table 10 is presented the pairwise comparison using the AHP methodology. The weights for each hazard were calculated based on Equation (9) and are presented in Table 10.

We have added a line (line 535) in the revised manuscript.

(7)  Line 610: This work is important to early warning systems in the wider are of the Aegean Sea, but in this article, it lacked detailed explanations on the potential early warning systems. Discussing the potential early warning system can highlight the novelty and contribution to the manuscript. Here I recommend two papers on tsunami early warning system that can be referred.

https://doi.org/10.1029/2022JB025153

Author’s response: Dear reviewer, thank you for your suggestions! Indeed, our study lacks in referring of early warning systems. We have integrated our text with the recommended reference in the discussion section.

Reviewer 2 Report

geohazards-2272923   REVIEW. 

This is an interesting paper on assessing the susceptibility of the Greek islands of Milos and Thira to various natural hazards and multi-hazards.  I think it can be published after revisions and corrections are made.  Some comments and questions: 

[1] In the captions of Figures 2 and 3, it might be worthwhile to mention that the labels in black boldface type (e.g., Sa, Pap, Ad, Kam, Rb) are abbreviations for the names of the beaches, ports, etc.

[2] Line 157: Can a reference be given for the MCDM? 

[3] Line 257: How are the weights (e.g., nS, nLC) determined?  Also, should the denominator in (1) be the sum of the weights (nS+nLC+nST+nV) instead of “4”?  That would be in accordance with the typical mathematical formula for an average.  And what about equation (12) (line 485).

[4] Line 277: Does n=0% indicate no slope (horizontal, M=10^0=1) and n=100% indicate a vertical slope (M=10^[100/30] = 2154)?  Also, in Table 3, the slope is in degrees, not percent. 

[5] Line 331, eq. (5): What are the units of A? 

[6] Line 481, Table 6: In my opinion, the caption for Table 6 would be clearer if it read as follows: “Percentages of low-lying coastal areas that have very low to very high susceptibility to landslides in Milos and Thira islands.”  The same goes for the remaining susceptibility tables (7-9).  But it’s up to the authors. 

[7] Line 540, Table 10: I tried to reproduce the figures in Table 10 using equation (9) for the weights w_i.  It seems to me the figures are not quite right.  I assume that the figures in the second-last column are the geometric means of the rows (why is it called “Sum(Ui)”?).  The table shows that the geometric mean for the first row is 1.8633.  But when I calculate it, I get (1x2x3x4)^(1/4) = 2.2134, which is quite different.  Similarly, I get, for the geometric means of the second, third and fourth rows, 1.3161, 0.75793, and 0.45067, respectively, which are significantly different from those in Table 10.  The sum of the means that I calculated is 4.73803.  Therefore, the first weight that I calculate is w_1 = 2.2134/4.73803 = 0.4672.  But your Table 10 has 0.4658, which is a bit different.  Similarly, the second, third and fourth weights that I calculated are w_2 = 0.2778, w_3 = 0.1600, w_4 = 0.0951.  These are all a bit different from the weights in Table 10 (although they are close).  Am I correct?  Are your figures wrong?  Or did I do something wrong?    If I am correct, then equation (13) would need to be corrected. 

[8] General question: Since landslides, soil erosion, etc. change the topography, would the weights for these hazards not change with time?  And how would that be incorporated into your analysis? 

Author Response

We appreciate the opportunity to submit the revised version of the paper "Multi-hazard susceptibility assessment using the Analytical Hierarchy Process in Coastal Regions of South Aegean Volcanic Arc Islands" for consideration by the GeoHazards Journal. We value the resources and time you invested in offering comments on our manuscript. We are also appreciative of the helpful comments and enhancements made to our paper. Please see below, for a detailed response to the reviewers’ comments and concerns.

This is an interesting paper on assessing the susceptibility of the Greek islands of Milos and Thira to various natural hazards and multi-hazards.  I think it can be published after revisions and corrections are made.  Some comments and questions: 

[1] In the captions of Figures 2 and 3, it might be worthwhile to mention that the labels in black boldface type (e.g., Sa, Pap, Ad, Kam, Rb) are abbreviations for the names of the beaches, ports, etc.

Author’s response: Dear reviewer, thank for pointing this out! Indeed, we should specify the abbreviations that presented in Figure 2 and 3. We revised accordingly.

[2] Line 157: Can a reference be given for the MCDM? 

Author’s response: Dear reviewer, the you for your suggestion. We have added a reference for the MCDM.

[3] Line 257: How are the weights (e.g., nS, nLC) determined?  Also, should the denominator in (1) be the sum of the weights (nS+nLC+nST+nV) instead of “4”?  That would be in accordance with the typical mathematical formula for an average.  And what about equation (12) (line 485).

Author’s response: We agree the explanation of the FFPI calculation could have been more descriptive. According to the FFPI methodology bibliography, the calculation of the Flash Flood Potential Index is calculated by multiplying each factor with their respective weight (ni), summarizing them and finally dividing them by the maximum number of factors considered in the torrential flood analysis. Additionally, in bibliography a variate of different techniques was used to calculate the weights, however in the case of our research all factors were considered as equal. This information has been added to the revised manuscript.

[4] Line 277: Does n=0% indicate no slope (horizontal, M=10^0=1) and n=100% indicate a vertical slope (M=10^[100/30] = 2154)?  Also, in Table 3, the slope is in degrees, not percent. 

Author’s response: We would like to note that M wasn’t used for the calculation of the slope gradient. Particulary the slope was calculated in degrees and then was reclassified. The manuscript has been revised accordingly.

[5] Line 331, eq. (5): What are the units of A? 

Author’s response: Thank you for pointing this out! The A computes soil loss in tons per hectare per year (t ha−1 y−1). We have included this clarification in the updated manuscript.

[6] Line 481, Table 6: In my opinion, the caption for Table 6 would be clearer if it read as follows: “Percentages of low-lying coastal areas that have very low to very high susceptibility to landslides in Milos and Thira islands.”  The same goes for the remaining susceptibility tables (7-9).  But it’s up to the authors. 

Author’s response: Dear reviewer, thank you for your guidance! We have changed the caption in Table 6-9 as you suggested.

[7] Line 540, Table 10: I tried to reproduce the figures in Table 10 using equation (9) for the weights w_i.  It seems to me the figures are not quite right.  I assume that the figures in the second-last column are the geometric means of the rows (why is it called “Sum(Ui)”?).  The table shows that the geometric mean for the first row is 1.8633.  But when I calculate it, I get (1x2x3x4)^(1/4) = 2.2134, which is quite different.  Similarly, I get, for the geometric means of the second, third and fourth rows, 1.3161, 0.75793, and 0.45067, respectively, which are significantly different from those in Table 10.  The sum of the means that I calculated is 4.73803.  Therefore, the first weight that I calculate is w_1 = 2.2134/4.73803 = 0.4672.  But your Table 10 has 0.4658, which is a bit different.  Similarly, the second, third and fourth weights that I calculated are w_2 = 0.2778, w_3 = 0.1600, w_4 = 0.0951.  These are all a bit different from the weights in Table 10 (although they are close).  Am I correct?  Are your figures wrong?  Or did I do something wrong?    If I am correct, then equation (13) would need to be corrected. 

Author’s response: Dear reviewer, thank you for highlighting this. We replaced the column Sum(Ui), which was the sum of each line of the normalized values of pairwise matrix in table 10 with the geometric mean based on the normalized values of the matrix. We have made the modifications as you mentioned in your comment. In the figure bellow, you can check our latest calculated weight values which are very close to your calculations.

[8] General question: Since landslides, soil erosion, etc. change the topography, would the weights for these hazards not change with time?  And how would that be incorporated into your analysis? 

Author’s response: Dear reviewer, thank you for your insightful question. Indeed, these specific weights have been selected based on the frequency occurrence of the analysed hazards. However, additional scenarios and different hazards can be adopted and further processed.  This work tries to introduce within the boundaries of the low-lying coastal a complex issue in terms of multi-hazard approach. Specifically, the weights have been produced using multi-criteria decision-making approach in order to balance and quantify the importance of each hazard. In the perspective of a new scenario based on other hypotheses, such as a sudden climate change or a severe geohazard in the widen territory, the weights of these hazards could be adjusted.
